# New Approach for Studying of Isoforms and High-Homology Proteins in Mammalian Cells

**DOI:** 10.3390/ijms241512153

**Published:** 2023-07-29

**Authors:** Nataliya V. Soshnikova, Yuriy P. Simonov, Alexey V. Feoktistov, Alvina I. Khamidullina, Margarita A. Yastrebova, Darya O. Bayramova, Victor V. Tatarskiy, Sofia G. Georgieva

**Affiliations:** 1Department of Transcription Factors, Engelhardt Institute of Molecular Biology, Russian Academy of Sciences, Vavilov St. 32, Moscow 119991, Russia; 2Center for Precision Genome Editing and Genetic Technologies for Biomedicine, Engelhardt Institute of Molecular Biology, Russian Academy of Sciences, Vavilov St. 32, Moscow 119991, Russia; 3Department of Molecular Oncobiology, Institute of Gene Biology, Russian Academy of Sciences, Vavilov St. 34/5, Moscow 119334, Russia

**Keywords:** protein isoforms, homology proteins, PHF10, pSLIK

## Abstract

In mammals, a large number of proteins are expressed as more than one isoform, resulting in the increased diversity of their proteome. Understanding the functions of isoforms is very important, since individual isoforms of the same protein can have oncogenic or pathogenic properties, or serve as disease markers. The high homology of isoforms with ubiquitous expression makes it difficult to study them. In this work, we propose a new approach for the study of protein isoforms in mammalian cells, which makes it possible to individually detect and investigate the functions of an individual isoform. The approach was developed to study the functions of isoforms of the PHF10 protein, a chromatin subunit of the PBAF remodeling complex. We demonstrated the possibility of induced simultaneous suppression of all endogenous PHF10 isoforms and the expression of a single recombinant FLAG-tagged isoform. For this purpose, we created constructs based on the pSLIK plasmid with a cloned cassette containing the recombinant gene of interest and miR30 with the corresponding shRNAs. The doxycycline-induced activation of the cassette allows on and off switching. Using this construct, we achieved the preferential expression of only one recombinant PHF10 isoform with a simultaneously reduced number of all endogenous isoforms. Our approach can be used to study the role of point mutations, the functions of individual domains and important sites, or to individually detect untagged isoforms with knockdown of all endogenous isoforms.

## 1. Introduction

The vast majority of eukaryotic genes, especially for mammals, code several transcriptional variants—gene isoforms. These isoforms are products of alternative splicing: the existence of alternative transcription start sites, termination sites, polyadenylation, and alternative inclusion of exons. In mammals, 95% of all genes code for more than one transcript, which differ in their exon structure, and 35% of genes code for more than one polypeptide [1].

Another source for protein isoforms is closely related or almost identical sequences, which are transcribed from different gene loci. Homologous proteins and protein isoforms, which are transcribed from different genes, can be interchangeable and ubiquitously expressed, as, for example, the family of structural actin proteins [2]. Such genes can appear because of gene duplication and are then fixed in the genome by natural selection. The interchangeability and duplication of functions gives the organism more chances to survive spontaneous mutations affecting essential genes. Some genes can perform different functions, despite high homology, as, for example, in the SWI/SNF remodeling complex, where the ATPase can be represented by one of the two highly homologous proteins—BRG1 or BRM1—with 75% homological sequences, which, nevertheless, are not interchangeable in embryonic development [3].

Previously, it was thought that protein isoforms have similar functions, but recent research demonstrated that it is not always the case. The existence of multiple isoforms that have major or little difference in their amino acid sequence can lead to substantial differences in domain organization and secondary structure. Structural differences allow the proteins to have different special functions, allowing proteins to function in changing environments and respond to intracellular and extracellular signals. For example, RET9 and RET51 are different isoforms of the receptor tyrosine kinase *RET* gene. They have different trafficking properties, interact with different adaptor proteins, and, as a result, induce distinct gene expression patterns, promote different levels of cell differentiation and transformation, and play unique roles in development [4].

Isoforms can be ubiquitous, tissue specific, or play an important role at certain stages of development, as in the formation of the brain, the heart, immune homeostasis, or sex determination [5,6,7,8]. The Reep6.1 isoform is only expressed in the adult retina and has rod-specific functions that cannot be substituted by the Reep6.2 canonical isoform. The deletion of the Reep6.1-specific exon gives a rod degeneration phenotype in knockout mice [9].

Defects in splicing and the incorrect expression of isoforms cause development pathologies, impair the functioning of the brain, heart, cognition, and others, and even accelerate aging [10,11,12,13,14,15]. hnRNPDL is an RNA-binding protein and transcriptional regulator expressed as three isoforms as a result of alternative splicing. Only one of them containing a specific exon forms amyloid filaments. Point mutations in its sequence are a cause of autosomal dominant limb–girdle muscular dystrophy D3 associated with impaired fibrillation [16].

Aberrant isoform expression is linked to oncological transformation [17,18,19,20]. For example, the A isoform of myosin 1C is increased in prostate cancer [21]. Different isoforms can have opposing roles in cancer—while BRD4-S is pro-oncogenic, BRD4-L has a tumor-suppressive role [22].

The investigation of individual protein isoforms and homologs, which are ubiquitously expressed, is complicated, as their functional effects can be masked or complemented by the expression of similar isoforms. Experiments where an isoform is recognized with antibodies are often hampered by the absence of specific antibodies against individual isoforms and the screening of one isoform by another. Similarly, it is difficult or even impossible to design suitable isoform-specific primers and shRNA. The closer isoform sequences are to each other, the more difficult they are to study individually. The extreme variant is the study of point mutations, as in the RAS GTPase family (NRAS, HRAS, KRAS), and their mutated variants, which are present in 30% of all tumors [23,24,25].

Here, we propose a methodological approach that allows the retaining of only the desired isoform by knocking down all endogenous isoforms and the simultaneous re-expression of the target protein. We tested this approach on isoforms of the PHF10 protein. To this end, we constructed a plasmid, based on the pSLIK vector, which allows the doxycycline knockdown of all endogenous isoforms with shRNA, processed from an miR30 cassette, and simultaneous overexpression of a recombinant Flag-tagged protein that is resistant to the knockdown. We tested this approach on all four isoforms of PHF10 in three cell lines. The advantages of this vector are the all-in-one design, in which all elements are contained in one plasmid; doxycycline-induced expression, which allows the knockdown and expression to be turned on and off; the option for lentiviral delivery in any type of cell line; and G418 selection (Figure 1).

Therefore, our approach allows the study of individual isoforms, protein homologs, and mutant variants in cells.

## 2. Results

### 2.1. Recombinant Mutated Isoforms of PHF10 Are Resistant to Knockdown

To prove the feasibility of the strategy to study individual isoforms, we used an example of a subunit of the PBAF complex—PHF10. The PHF10 protein is expressed as four isoforms in mammalian cells (Figure 2A). Isoforms are different transcripts—products of one gene—*PHF10*, which have alternative 5′ and 3′ exons, being transcribed from alternative promoters and possessing alternative termination sites. The isoforms differ from each other in their N- and C-termini and are extensively phosphorylated [26]. In most cell lines, all isoforms are expressed simultaneously. To study the functions of each isoform, it would be beneficial to have cell lines where only one isoform would be expressed, while others would be suppressed. Short double-strand RNA oligonucleotides are used to downregulate protein expression in mammalian cells, either through the transfection of siRNA or the expression of shRNA from a plasmid. Previously, we determined two 19-nt sequences in the *PHF10* gene, which, when targeted by an siRNA or by shRNA from a pGPV vector, would lead to the effective knockdown of all four isoforms of PHF10 [27,28]. To protect the recombinant isoforms from the knockdown, we introduced silent mutations into the PHF10 sequences, designated as PHF10-RI and PHF10-RII (Figure 2B). Isoforms that are resistant to both shRNA were called PHF10-R. To make sure that the mutated isoforms would have stable expression on the background of suppression of the endogenous isoforms, we transfected HEK293T cells with two plasmids: pGPV-PHF10-shI/shII (KD-PHF10) for the knockdown, and Fl-PHF10-wt and Fl-PHF10-R in the pcDNA vector for overexpression (Figure 2C).

Staining with antibodies, which recognize all endogenous isoforms of PHF10, showed a significant downregulation of all endogenous isoforms and the recombinant Fl-PHF10-wt by the KD-PHF10. The recombinant isoforms Fl-PHF10-R had a comparable expression with PHF10-wt in cells without the knockdown (HEK293T), but were not targeted by RNA interference in the cells with the knockdown, compared to PHF10-wt, which were fully downregulated (based on anti-Flag staining) (Figure 2C). Therefore, in HEK293T, we achieved a full knockdown of all endogenous isoforms and the overexpression of recombinant isoforms, not recognized byPHF10-shI and PHF10-shII, allowing for the expression of one isoform out of the four possible.

### 2.2. pSLIK Vector Provides Effective Induced Knockdown and Simultaneous Overexpression of PHF10 Isoforms

Previously, the D.C. Fraser group had constructed a pSLIK (single lentivector for inducible knockdown) vector, which allows the simultaneous efficient knockdown of endogenous proteins with shRNA, expressed from a CMV promoter by RNA-pol II as miR30, and then processed by endogenous enzymes [29]. A TRE element is placed before the CMV promoter, which binds to the rtTA3 protein when doxycycline (Dox) is added, activating the transcription from CMV (TET-ON system) [30]. The plasmid contains the rtTA3 activator itself, under a constant ubiquitin promoter (UBC), which works in all mammalian cells. The vector also contains elements that allow its inclusion in lentiviral particles, which provides for the efficient introduction of the vector into mammalian cells by transduction (Figure 3A,B).

We tested the ability of this vector to efficiently knock down endogenous isoforms of PHF10 and overexpress Flag-tagged, knockdown-resistant PHF10 isoforms in the other cell line—HEK293.

To test the knockdown, we cloned PHF10-shI and PHF10-shII into miR30 cassettes, producing the pSLIK-(shI + shII) vector, transduced HEK293 with lentiviral particles and selected cells on G418. Before selection, Dox was added for 48 h. As a control, HEK293 was transduced with lentiviral particles based on plasmids pGPV-shI and pGPV-shII, providing a constant knockdown of PHF10 isoforms, and selected on puromycin. The use of two shRNA against the PHF10 plasmid allows for the more efficient downregulation of endogenous isoforms. The induced knockdown with the pSLIK-(shI + shII)-based lentivirus was comparable to constant knockdown (Figure 3C).

To test the overexpression and resistance to the knockdown, PHF10 (PHF10i—any isoform of PHF10) was cloned into a pSLIK vector (without the miR30 cassette) (Figure 3B) and then transduced into HEK293 with pGPV-shI and pGPV-shII in equal proportions. The cells were selected on both G418 and puromycin and then induced with Dox for 48 h. We detected a drop in the level of all endogenous PHF10 isoforms compared with the control HEK293 and an increase in the expression of recombinant isoforms when Dox (1ug/mL) was added (Figure 3D). The difference in the expression of Fl-PHF10-Pl/Sl and Fl-PHF10-Ps/Ss isoforms can be explained by the different stability of the isoforms.

Thus, the pSLIK vector allows the efficient induced downregulation of endogenous isoforms and the expression recombinant isoforms of PHF10.

### 2.3. Activation of GOI Transcription and Knock down with pSLIK Enables Expression of a Single PHF10 Isoform in Different Cell Lines

Subsequently, we assembled a cassette with the induced expression of recombinant PHF10 and two miR30 with shI and shII in one pSLIK (miR30 cassettes are contained in the 3′ untranslated region of the transcript, and shRNAs are processed from there by the cell’s RNA interference machinery), which was transduced in HEK293 to obtain stable lines after G418 selection (Figure 4A). After activation with Dox for 48 h, we detected a significant increase in the protein levels of Fl-PHF10i and a reduction in endogenous PHF10 (Figure 4B).

To determine the levels of endogenous and recombinant PHF10 quantitatively, we used real-time PCR with primers that recognize all PHF10 isoforms—PHF10-total; primers that recognize isoforms containing DPF domains (for PHF10-P); and primers for isoforms from which DPF domains are absent—PHF10-S. In the selected stable HEK293 cell lines, the induction of all recombinant isoforms increased four-fold compared to the endogenous isoforms without induction (Figure 4C; PHF10-total). The expression of individual recombinant isoforms increased 7–8-fold in the stable cell lines (Figure 4C; PHF10-P и PHF10-S). The activation of recombinant PHF10-Pl and PHF10-Ps expression led to the downregulation of endogenous PHF10-S, and, alternatively, in HEK293 with the expression of recombinant PHF10-Sl и PHF10-Ss, the level of PHF10-P was lower, confirming the activity of endogenous isoforms by PHF10 shI and shII (Figure 4C).

We also tested our constructs in the A375 melanoma cell line. We detected an increase in the expression of recombinant isoforms and the downregulation of endogenous isoforms in all selected A375 sublines, both by Western blot and real-time PCR (Figure 4D,E).

In immortalized human fibroblasts (HF), the Fl-PHF10i pSLIK constructs had a minor increase in expression without Dox (also known as “leakage”) (Figure 4F, recombinant isoforms before induction are denoted with arrows), but the activation of PHF10i expression increased transcription 30-fold (Figure 4G). The downregulation of endogenous PHF10 isoforms by the shRNA was lower than in HEK293 and A375. We suggest that this effect is explained by the molecular context of fibroblasts, where the expression from the TRE-CMV promoter is more efficient with or without the addition of Dox. These particular qualities of individual cell lines should be considered when our approach is employed.

Thus, our approach provides the ability to simultaneously induce one isoform of PHF10 and knockdown expression of its endogenous isoforms in the same cell, allowing the study of the functions of PHF10 isoforms in a variety of mammalian cells.

### 2.4. Activation of PHF10 Isoform Expression by Dox Is Dose Dependent

Next, we tested whether the transcription of Dox is dose dependent, using A375 Fl-PHF10i stable cell lines. A375 cells with pSLIK-Fl-PHF10-Pl(shI + shII) were incubated with different concentrations of Dox, ranging from 0.01 to 1 μg/mL for 48 h, after which the cells were fixed and stained with antibodies against the Flag epitope. We detected an increase in the number and intensity of fluorescent cells with the increase in the Dox concentration (Figure 5A).

Similarly, we detected a dose-dependent gradual increase in recombinant isoforms by Western blot in A375 PHF10i cell lines. However, the efficient knockdown of endogenous isoforms was only achieved at the 1 μg/mL dose of Dox in the medium (Figure 5B).

Thus, by varying the concentration of Dox in the medium, it is possible to regulate the intensity of recombinant isoform expression in cell lines with the pSLIK construct.

### 2.5. Detection of Individual Isoforms with Antibodies against All PHF10 Isoforms

If individual isoforms of a protein cannot be distinguished by antibodies, the only possible solution is to use overexpressed isoforms with various N- and C-terminal tags. We tested whether, using our approach, it is possible to distinguish individual PHF10 isoforms with antibodies against all isoforms. Previously, we had shown that individual isoforms have different nuclear and cytoplasmic localization in cells [26,31]. We activated stable HEK293 cell lines containing pSLIK-Fl-PHF10i (shI + shII) with doxycycline for 48 h and then stained the cells with antibodies against total PHF10 (Figure 6). Recombinant PHF10-Pl and PHF10-Sl isoforms were only localized in the nucleus, while PHF10-Ps and PHF10-Ss were also present in the cytoplasm, similarly with their endogenous counterparts (Figure 6).

Thus, our approach allows us to detect recombinant isoforms, with antibodies against the total protein, when tags can compromise their functions.

## 3. Discussion

In the present work, we have developed a strategy for studying the individual functions of highly homologous ubiquitously expressed proteins or protein isoforms in mammalian cells. We used a pSLIK vector with a cloned cassette containing the recombinant gene of interest and miR30 with corresponding shRNA, enabling us to induce the overexpression of a recombinant isoform and knockdown of all endogenous isoforms. By using a single plasmid construct, knockdown and overexpression occur simultaneously in each cell carrying this plasmid. Viral elements in the plasmid allow the generation of lentiviruses based on it for fast and efficient delivery into a variety of cell lines.

Using the example of closely related protein isoforms of PHF10, we demonstrated that upon the transcriptional activation of constructs by doxycycline in HEK293, A375, and HF cells, only one recombinant isoform is expressed in the cells after 48 h. Often, in the case of the conventional overexpression of a mutated protein in cells, it does not replace the full-length endogenous form because of reduced functionality, and the mutation effect is, therefore, not sufficiently manifested. By using the pSLIK plasmid for overexpression of the gene of interest on the background of the knockdown, the endogenous proteins are replaced by recombinant ones in the processes and signaling pathways involving the studied proteins. This allows the study of functions of individual domains and important sites.

Our approach can be used to study the role of point mutations by cloning the corresponding recombinant sequence as the gene of interest. The induction of the mutated protein using Dox allows for the simulation of mutations in important proteins and, therefore, the study of processes leading to oncological transformation.

Additionally, using our constructs of PHF10 isoforms in stable HEK293 cell lines, we have shown that with the help of pSLIK-Fl-PHF10i (shI + shII) constructs, it is possible to detect individual recombinant isoforms using antibodies that recognize all endogenous isoforms. Overexpression and knockdown upon activation of the constructs are sufficient to create an excess of the individual recombinant isoform over the endogenous ones, which can then be localized using antibodies that do not distinguish between isoforms. This strategy allows the study of high-homology proteins using such antibodies in cases where additional amino acids at the ends of the protein negatively affect its functionality, and it is not possible to use additional tags at the N- or C-termini of the proteins.

During our validation of the approach for studying the functions of individual isoforms, we encountered low spontaneous expression of isoforms from the inducible TRE-CMV promoter without the addition of Dox in immortalized human fibroblasts. We explained this by a specific molecular context that allows RNA polymerase II to transcribe from the CMV promoter without activation or the rtTA3 activator binding to the TRE element. Promoter leakage can also occur if the FBS added to the culture medium contains traces of tetracycline antibiotics. Sometimes, this effect can be undesirable, and in such cases, it is necessary to carefully select reagents (free of tetracycline antibiotic traces) or use appropriate cell lines.

## 4. Materials and Methods

### 4.1. Cloning

Plasmids with the individual PHF10 isoforms pcDNA-Fl-PHF10-Pl/Sl/Ps/Ss were generated by our group previously [26]. Double-silence mutants resistant against PHF10-shI and PHF10-shII were performed as subsequent PCR with for: 5′-CTGCTCTCAGATCAGACGAAGTGATTGAT and rev: 5′-CACTTCGTCTGATCTGAGAGCAGTTAAGCCTAG primers (against PHF10-shI) and for: 5′-AAAGTGTCAAGCTATCCGGTGGCTCTCATC and rev: 5′-CACCGGATAGCTTGACACTTTTGTTCGCTC primers (against PHF10-shII) on the hole plasmids. Then, the PCR mixture was treated by the DpnI restriction enzyme (Thermo Fisher Scientific, Waltham, MA, USA) for matrix degradation.

For pGPV-PHF10-shI (for: 5′-GATCC*CAGCATTGCGCAGTGATGA*TTCAAGAGA*TCATCACTGCGCAATGCTG*TTTTTG and rev: 5′-AATTCAAAAA*CAGCATTGCGCAGTGATGA*TCTCTTGAATCATCACTGCGCAATGCTGG) and pGPV-PHF10-shII (for: 5′-GATCC*AGGTCAGTTCTTACCCAGT*TTCAAGAGA*ACTGGGTAAGAACTGACCT*TTTTTG and rev: 5′-AATTCAAAAAAGGTCAGTTCTTACCCAGTTCTCTTGAAACTGGGTAAGAACTGACCTG)-mediated knockdown, the sequences were cloned in a pGPV vector (Evrogen, Moscow, Russia) previously treated with BamHI and EcoRI restriction enzymes (Thermo Fischer Scientific, Waltham, MA, USA) for generating the short hairpin.

Two miR30 cassettes with PHF10-shI and PHF10-shII sequences were synthesized (Evrogen, Moscow, Russia) and tandem cloned in pBlueSK plasmid. Then, these tandem cassettes and two parts of pSLIK-Neo (the first from UBC promoter to SV40 promoter, the second from SV40 to CMV promoter) were PCRed and assembled with NEBuilder (NEB, Ipswich, MA, USA) in one pSLIK-PHF10 (shI + shII) plasmid. pSLIK-Neo was a gift from Iain Fraser (Addgene plasmid # 25735, Watertown, MA, USA) [29].

Individual resistant Fl-PHF10-Pl/Sl/Ps/Ss isoforms also were PCRed from pcDNA plasmid and inserted between the CMV and UBC promoters of pSLIK with NEBuilder instead of *GFP* gene.

Resistant Fl-PHF10-Pl/Sl/Ps/Ss isoforms were cloned upstream in pBlueSK-PHF10 (shI + II)-intermediate plasmid, PCRed, and assembled in pSLIK with NEBuilder as pSLIK-Fl-PHF10i (shI + shII).

### 4.2. Cells

HEK293T (CRL-3216), HEK293 (CRL-1573), A375 (CRL-1619) (all from ATCC collection), and immortalized human fibroblasts (HF) (obtained via transduction of hTERT gene in human skin fibroblasts in Engelhardt Institute of Molecular Biology of RAS, Moscow, Russia; gift from E. Dachinimaev [32]) cells were grown in DMEM medium supplemented 10% FBS (HyClone, Logan, UT, USA), 2 mM L-glutamine, and 1× Ampicillin/Streptomycin antibiotic (HyClone, Logan, UT, USA) at 37 °C, 5% CO_2_. All cell lines were routinely tested for Mycoplasma contamination by DAPI staining.

### 4.3. Transfection and Lentivirus Transduction

Transient transfection of HEK293T cells was performed with pcDNA-Fl-PHF10-Pl/Sl/Ps/Ss, pGPV-PHF10-shI/shII, or pSLIK-Fl-PHF10i (shI + shII) with pCMV-VSV-G (Addgene #8454 [33]) and pCMV-dR8.2 (Addgene #8455 [33]) plasmids. The plasmids were mixed with polyethylenimine (1 mg/mL, Polysciences, Warrington, PA, USA) as 1:2 (*w*/*v*) in serum-free media Opti-MEM (Thermo Fisher Scientific, Waltham, MA, USA). The mixtures were incubated for 15 min at room temperature and added to cells grown to 50% confluence. We used 10μg plasmids for transfection in a 60 mm Petri dish and 30 μg plasmids in a 90 mm Petri dish. Overnight, the media was changed. The cells were harvested 48 h post transfection.

To obtain lentivirus particles, pSLIK-Fl-PHF10i (shI + shII) plasmids were co-transfected into HEK293T cells together with VSV-G and pCMV-dR8.2 vectors (10 μg of each plasmid per 90 mm Petri dish). Supernatants with viral particles were collected on the second day after transfection, filtered through a 0.22 μm membrane, and then 10 mL was added to the appropriate cells. The next day, the media was replaced with the media containing 0.5 μg/mL G418. For stable sublines, the selection continued for 10 days.

### 4.4. Western Blotting

Precipitated proteins were eluted with for immunoblotting, and cells were lysed in RIPA buffer (50 mM Tris-HCl pH 7.4; 1% NP-40; 0.5% Na deoxycholate; 0.1% SDS; 150 mM NaCl; 2 mM EDTA; PIC and PhIC) and centrifuged at 12,000 rpm, 4 °C. The protein concentration was measured using a Qubit protein assay kit (Thermo Fisher Scientific, Waltham, MA, USA) and mixed with 4× Laemmli buffer (200 mM Tris–HCl pH 6.8; 4% SDS; 40% glycerol; 0.01% bromophenol blue; 100 mM DTT) and then boiled for 10 min. After electrophoresis, the proteins were transferred to the nitrocellulose membrane (Bio-Rad, Hercules, CA, USA) and stained with 5% milk (Cell Signaling Technology, Danvers, MA, USA) in PBS. Endogenous PHF10 isoforms were detected by anti-PHF10 antibodies described previously [28]. These polyclonal antibodies were generated in our laboratory by immunizing rabbits with the His-tagged polypeptide corresponding 89–370 amino acids (NP_060758.2) contained in all four PHF10 isoforms. The M2 clone (Sigma-Aldrich, Saint Louis, MO, USA) was used as anti-FLAG antibodies. Anti-Tubulin (#2148) and anti-GAPDH (14G10) antibodies were from Cell Signaling. HRP-conjugated anti-rabbit IgG and HRP-conjugated anti-mouse goat IgG were from DHGB.

### 4.5. Gene Expression Analysis

RNA was isolated from 3 × 10^6^ cells using TriReagent (MRC, Beverly Hills, CA, USA) according to the manufacturer’s protocol. Then, 3 μg RNA was used for cDNA synthesis with oligo(dT) primer (Evrogen, Moscow, Russia) and MMLV reverse transcriptase supplemented with RiboLock RNAse inhibitor (both Thermo Fisher Scientific, Waltham, MA, USA). We used PCR primers for *PHF10-P* (5′-CCAGGGAAGACAGAAATCAAAAGAC and 5′-CCATTGTCATATCCAGGCAAGAAGG), *PHF10-S* (5′-CCAGGGAAGACAGAAATCAAAAGAC and 5′-CAGGGGCTTTTTTCTTCTACCTTG), *PHF10-total* (5′-CCGGGAACGCATGGAAGAAAG and 5′-CACCATCACTGTCTAGAGCAGGGAGC) and *RPLP0* (5′-ACTGGAGACAAAGTGGGAGCC and 5′-CAGACACTGGCAACATTGCG) housekeeping gene for value normalization. At least three independent experiments were performed; values are mean ± SD. Statistical analysis was performed using a one-way ANOVA with Dunnett’s multiple comparison test using GraphPad Prism 6 software. *p* values < 0.05 were considered significant.

### 4.6. Immunostaining

For immunostaining, cells were grown on cover glasses under previously described conditions. After 48 h of Doxycycline induction, the cells were fixed with 3.7% formaldehyde (Sigma-Aldrich, Saint Louis, MO, USA) permeabilized with 0.2% Triton-X100 and immunostained with anti-PHF10 or anti-Flag antibodies (both 1:100) for 1 h at RT, washed three times with PBS, and stained with the secondary anti-rabbit or anti-mouse Alexa-488 Fluor-conjugated antibodies (1:200) (both Invitrogen, Carlbad, CA, USA) for 1 h at room temperature (RT). Nuclei were stained by DAPI (600 nM for 30 s). The stained preparations on glass slides were mounted in a mounting medium (Vector Laboratories, Newark, CA, USA) and examined under the DMR/HC5 fluorescent microscope (Leica, Wetzlar, Germany) with an HCX PZ Fluotar × 100/1.3 objective lens. Microscopic images were taken with a Leica DC350 F digital camera and analyzed with ImageJ software 1.53k.

## Figures and Tables

**Figure 1 ijms-24-12153-f001:**
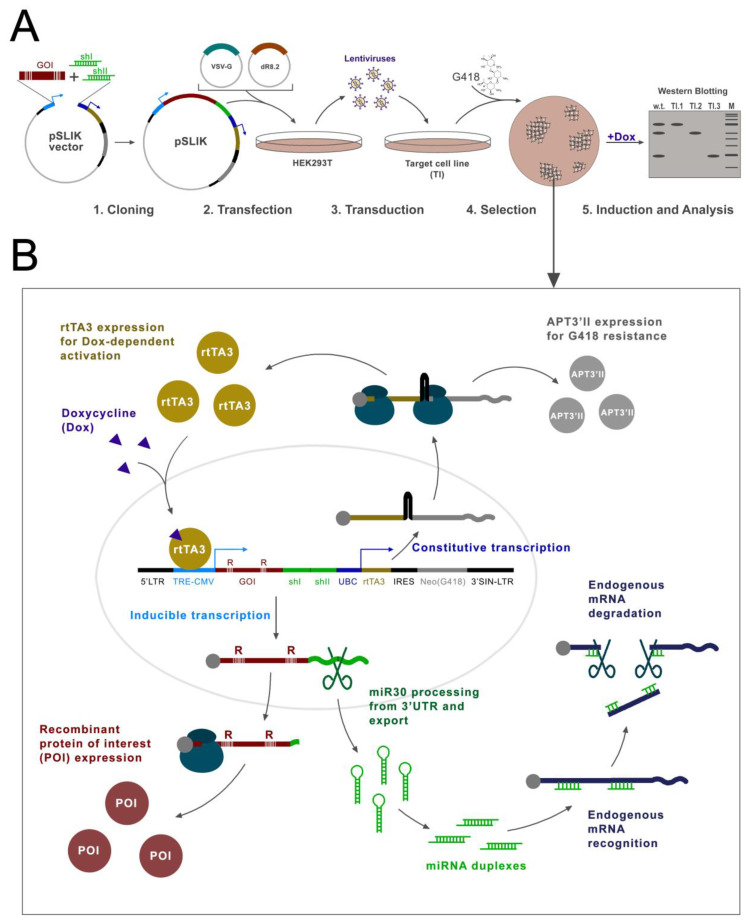
Scheme illustrating approach for studying individual isoforms and high-homology protein function. (**A**) Steps for generation of cell lines with pSLIK. (**B**) Scheme of different pSLIK elements activity. Black boxes 5′LTR and 3′SIN-LTR—elements for lentiviral assembly; blue boxes TRE CMV—promoter and TRE element for rtTA3 binding; green boxes miR30-shI/II—cassettes for shRNA cloning; red box GOI—gene of interest; red “R” letters and white vertical strips—silence mutations in sequence of GOI; green box with shI or shII—miR30-shI or miR30-shII; dark blue box UBC—promoter of *UBC* gene for constitutive transcription of activator rtTA3 (dark yellow box) and Neo gene of selection on G418 (gray) divided by IRES element (black box). Large gray empty circle depicted nucleus. More details are described in the text.

**Figure 2 ijms-24-12153-f002:**
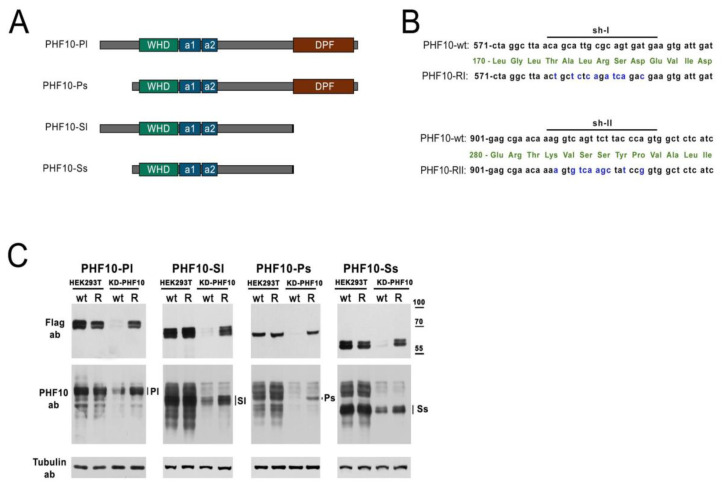
Isoforms of PHF10 carrying silent mutations in the sequences recognized by PHF10-shI and PHF10-shII are not subjected to knockdown in HEK293T cells. (**A**) Schematic representation of PHF10 protein isoforms. The green box represents the wing helix domain (WHD), the two blue boxes represent alpha helices (a1 and a2), and the dark red box represents the DPF domains of PHF10-Pl and PHF10-Ps isoforms. (**B**) Nucleotide sequences of PHF10-Pl mRNA isoform (starting from the ATG codon) in the wild-type PHF10-wt variant, the mutated forms PHF10-RI and PHF10-RII (dark blue letters), and their corresponding amino acid sequences (green letters). Extended sequences corresponding to the recognized parts of PHF10-shI and PHF10-shII are marked above with a line. (**C**) Western blotting of cells overexpressing recombinant wild-type (wt) and resistant (R) Fl-PHF10-Pl/Sl/Ps/Ss with (KD-PHF10) and without (HEK293T) knockdown of endogenous PHF10 isoforms. Expression of recombinant isoforms was detected using antibodies against the Flag epitope, and expression of both endogenous and recombinant isoforms was detected using antibodies against all PHF10 isoforms. Tubulin staining was used as a loading control.

**Figure 3 ijms-24-12153-f003:**
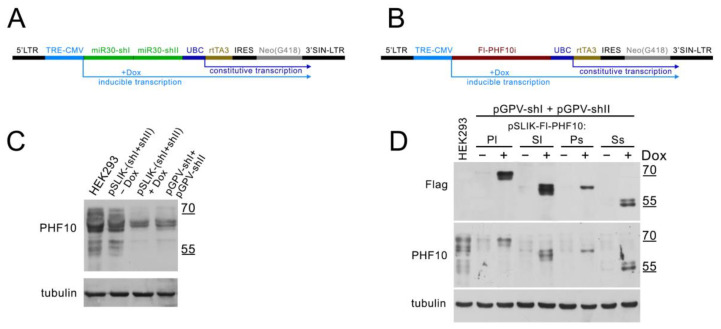
Efficient inducible knockdown and overexpression of recombinant PHF10 isoforms are achieved using the pSLIK plasmid. (**A**,**B**) Diagrams of constructs for knockdown (**A**) and overexpression (**B**) of PHF10 isoforms in cells (PHF10i—any of the four isoforms PHF10-Pl/Ps/Sl/Ss). Black boxes 5′LTR and 3′SIN-LTR—elements for lentiviral assembly; blue boxes TRE CMV—promoter and TRE element for rtTA3 binding; green boxes miR30-shI/II—cassettes for shRNA cloning; red box Fl-PHF10i—any of four Flag-tagged PHF10 isoforms resistant to shI/II; dark blue box UBC—promoter of *UBC* gene for constitutive transcription of activator rtTA3 (dark yellow box) and Neo gene of selection on G418 (gray) divided by IRES element (black box). (**C**) Western blotting analysis of wild-type HEK293 cells and cells with PHF10 isoforms knockdown by pSLIK-(shI + shII) activated (+Dox) and non-activated (−Dox) by Dox and by two pGPV-shI/II vectors. (**D**) Western blot analysis of wild-type HEK293 cells and cells with simultaneous knockdown by pGPV-shI + pGPV-shII plasmids and overexpression of resistant recombinant Fl-PHF10 isoforms (Pl, Sl, Ps, Ss) from pSLIK vectors with and without activation by Doxycycline (+/− Dox). Overexpressed PHF10 isoforms were detected by anti-Flag antibodies. In (**C**,**D**), all PHF10 isoforms in cells detected by anti-PHF10 antibodies and tubulin detection used as loading.

**Figure 4 ijms-24-12153-f004:**
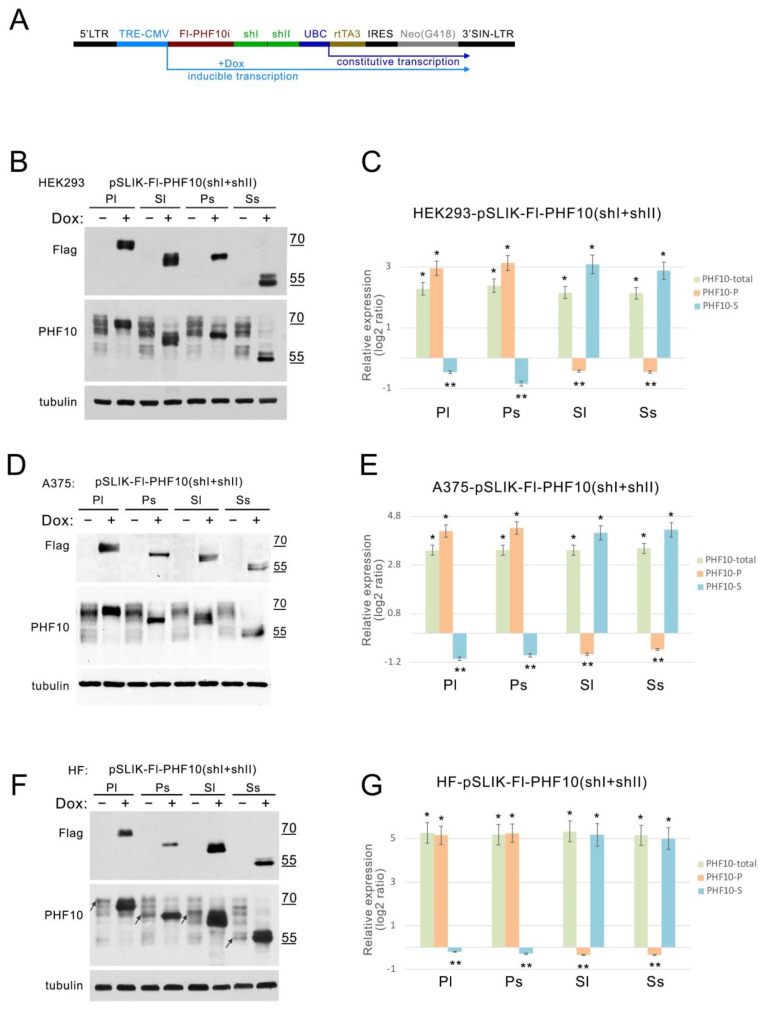
Induced expression from the pSLIK plasmid allows the preservation of a single PHF10 isoform in various cell lines. (**A**) Diagram of the construct for simultaneous inducible overexpression of recombinant isoforms and knockdown of endogenous PHF10 isoforms. See Figure 3 description for the details of the construct elements. Western blotting was performed on stable cell lines HEK293 (**B**), A375 (**D**), and HF (**F**) with or without (+/− Dox) activation of pSLIK-Fl-PHF10(shI + shII) constructs, where PHF10 represents one of the isoforms Pl, Ps, Sl, or Ss. Overexpressed PHF10 isoforms were detected by anti-Flag antibodies. All PHF10 isoforms in cells detected by anti-PHF10 antibodies and tubulin detection was used as loading. Expression of PHF10-P, PHF10-S, and total PHF10 level in the generated stable cell lines HEK293 (**C**), A375 (**E**), and HF (**G**) upon Dox activation. The values represent the mean ± SD from three independent experiments and were normalized to *RPLP0* expression. The Y-axis indicates the log2 of the RNA fold change. Fold induction, compared to control. *—*p* < 0.001, **—*p* < 0.01, (one-way ANOVA with Dunnett’s multiple comparison test).

**Figure 5 ijms-24-12153-f005:**
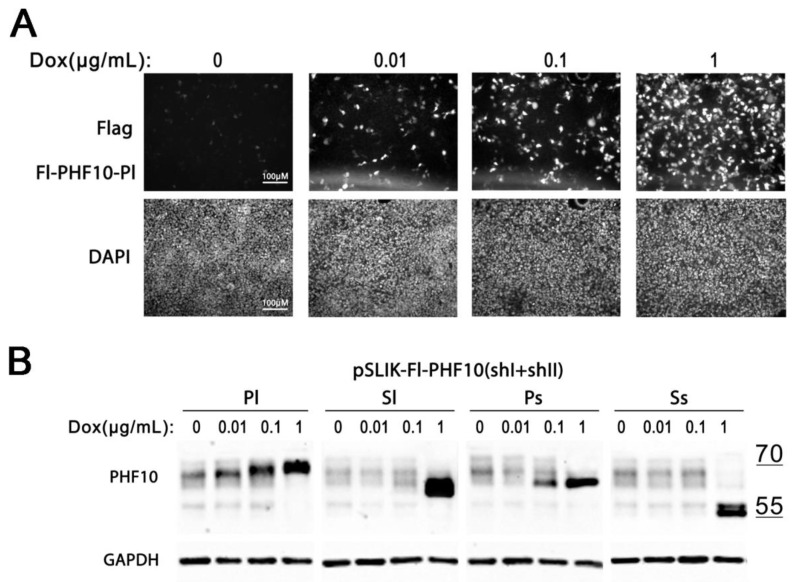
Activation of pSLIK-Fl-PHF10i(shI + shII) is dependent on Dox concentration in the medium. (**A**) Immunostaining of Fl-PHF10-Pl isoform in A375 cells upon activation with different concentrations of Dox in the medium. DAPI staining is provided to visualize the density of scattered cells. (**B**) Western blot analysis of the expression of recombinant-resistant isoforms Fl-PHF10-Pl/Sl/Ps/Ss and knockdown of endogenous isoforms upon activation with various concentrations of Dox (0.01, 0.1, and 1 μg/mL) using the corresponding pSLIK constructs. Antibodies against GAPDH were used for loading control.

**Figure 6 ijms-24-12153-f006:**
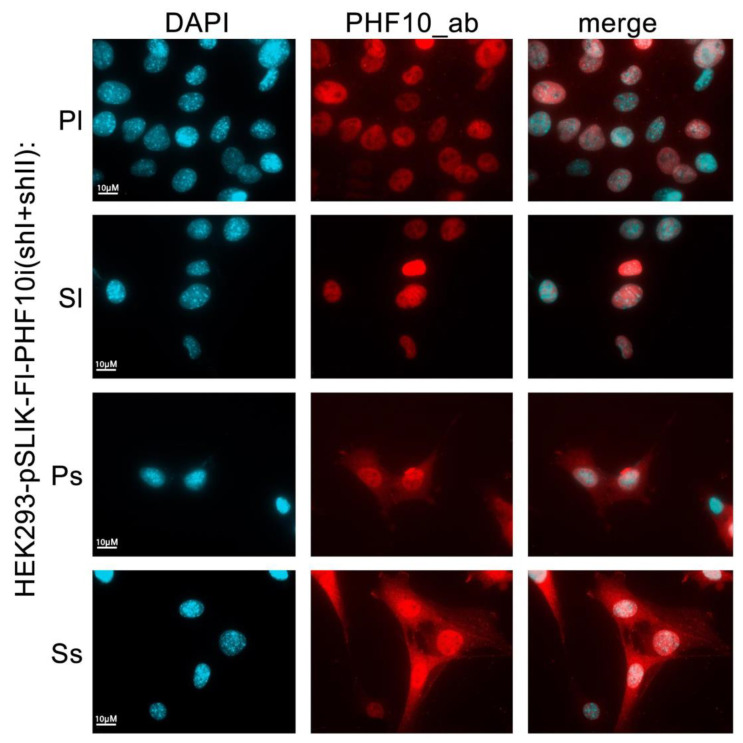
Immunostaining of recombinant PHF10 isoforms in HEK293 cells using antibodies against all PHF10 isoforms (PHF10_ab). DAPI was used to stain DNA in the nuclei. In the merged image, the localization of recombinant isoforms is detected: nuclear for PHF10-Pl/Sl and nuclear-cytoplasmic for PHF10-Ps/Ss.

## Data Availability

New data were not created.

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
