# Peer review of "New Approach for Studying of Isoforms and High-Homology Proteins in Mammalian Cells"

_ijms, 2023, doi:10.3390/ijms241512153_

Round 1

Reviewer 1 Report

Please see the attached PDF

Author Response

We thank the reviewers for their time and careful consideration of our manuscript.

We are grateful for generally positive evaluation of our work by the Referees and for their comments. We believe that we have made every effort to accommodate all their criticisms.

  1. In line 107, authors say, “Staining with antibodies, which recognize all endogenous isoforms of [.....]”

Are these monoclonal antibodies? What visualizing technique do authors use? Please clarify this in the materials and methods section.

- We previously generated these antibodies in our lab and tested them for detection of all four PHF10 isoforms on Western Blot and validated specificity with PHF10 knockdown. We published these results in article “Mammalian cells contain two functionally distinct PBAF complexes incorporating different isoforms of PHF10 signature subunit.” Brechalov AV, Georgieva SG, Soshnikova NV. Cell Cycle. 2014;13: 1970–1979.

We added the next clarify sentence in material and methods:

"These polyclonal antibodies were generated in our laboratory by immunizing rabbits with the His-tagged polypeptide corresponding 89-370 aminoacids (NP_060758.2) contained in all four PHF10 isoforms."

  1. Throughout the article, authors mention “HEK293” to refer to the cell line that they use. Please clarify if this is the same HEK293T cell line?

- We used two different lines HEK293T (ATCC CRL-3216) and HEK293 (ATCC CRL-1573). We clarified this in the text.

  1. Please doublecheck the text very carefully and correct all the chemical formulars. For example, in line 340 – CO2 should be CO2

- We corrected all formulars, misprints and other errors.

Reviewer 2 Report

The authors provide a new approach for study of isoforms and high-homology proteins in mammalian cells. The manuscript includes a lot of important details about pSLIK vector, enabling to induce overexpression of a recombinant isoform and knockdown of all endogenous isoforms. By using a single plasmid construct, knockdown and overexpression occur simultaneously in each cell carrying this plasmid.

Some details should be reorganized to clarity

My main suggestions are as follows:

There are a lot of grammar and  tense issues throughout the manuscript.  There are a lot of words with the lack of letter or space.

Figures should be presented near the text section describing them.

A list of abbreviations used should be added for example: GOI

The results presented are interesting, and the manuscript should be of interest to the readership of the International Journal of Molecular Sciences.

Author Response

We thank the reviewers for their time and careful consideration of our manuscript. We are grateful for generally positive evaluation of our work by the Referees and for their comments. We believe that we have made every effort to accommodate all their criticisms.

The authors provide a new approach for study of isoforms and high-homology proteins in mammalian cells. The manuscript includes a lot of important details about pSLIK vector, enabling to induce overexpression of a recombinant isoform and knockdown of all endogenous isoforms. By using a single plasmid construct, knockdown and overexpression occur simultaneously in each cell carrying this plasmid. 

Some details should be reorganized to clarity

My main suggestions are as follows:

1. There are a lot of grammar and  tense issues throughout the manuscript.  There are a lot of words with the lack of letter or space. 

- We corrected grammar, tense, and other mistakes in the manuscript.

2. Figures should be presented near the text section describing them.

- We transferred figures to the suitable sections.

3. A list of abbreviations used should be added for example: GOI

- All abbreviations are now deciphered in the text

Reviewer 3 Report

pro-2 teins in mammalian cells” submitted by Soshnikova and coworkers reports a novel approach for the study the functions of protein isoforms in mammalian cells. In particular, the authors have developed a construct based on the pSLIK plasmid in which they cloned the gene of interested together with miR30 and the corresponding shRNAs. Then the authors used doxycycline to induce the activation/inactivation of the expression and at the same time achieving the preferential expression of only one recombinant isoform variant. In this study, the authors used PHF10 protein as a model system. Taken together, the manuscript is well presented, organized and the topic reviewed is interesting for the field. However, I recommend the incorporation of additional data to improve the final quality of the work.

Minor and major comments:

1.     In the introduction section, the authors mention various examples of protein isoforms with distinct functions. However, it would be beneficial to include other relevant examples as well. For instance, hnRNPDL isoforms display distinct locations and splicing regulation functions (Nat. Comms. 14, 239, 2023). RET isoforms showed distinct trafficking properties (Mol Biol Cell. 23(19): 3838–3850, 2012). Reep6 has distinct functions in the retina (Hum Mol Genet. 13;30(21):1907-1918, 2021). These and other relevant case examples could be included in this section.

2.     I would suggest incorporating a schematic diagram of the approach at the end of the introduction section. This diagram can provide a general overview of how the different elements are inserted in the construct, how the plasmid is transferred to the cells, how it is selected, etc. This would help readers easily understand the proposed approach.

3.     Please include statistics in the different plots in Fig. 3

4.     It would be interesting if the authors could provide results of their approach for a different protein system, distinct from PHF10.

English is fine, only minor corrections are needed.

Author Response

We thank the reviewers for their time and careful consideration of our manuscript. We are grateful for generally positive evaluation of our work by the Referees and for their comments. We believe that we have made every effort to accommodate all their criticisms.

Minor and major comments:

  1. In the introduction section, the authors mention various examples of protein isoforms with distinct functions. However, it would be beneficial to include other relevant examples as well. For instance, hnRNPDL isoforms display distinct locations and splicing regulation functions (Nat. Comms. 14, 239, 2023). RET isoforms showed distinct trafficking properties (Mol Biol Cell. 23(19): 3838–3850, 2012). Reep6 has distinct functions in the retina (Hum Mol Genet. 13;30(21):1907-1918, 2021). These and other relevant case examples could be included in this section.

- We included more examples in the introduction.

  1. I would suggest incorporating a schematic diagram of the approach at the end of the introduction section. This diagram can provide a general overview of how the different elements are inserted in the construct, how the plasmid is transferred to the cells, how it is selected, etc. This would help readers easily understand the proposed approach.

- We inserted a diagram of our approach and how different elements in pSLIK work (Figure 1 in updated version).

  1. Please include statistics in the different plots in Fig. 3

- We included statistics in Fig.3 (Fig.4 in updated manuscript version).

  1. It would be interesting if the authors could provide results of their approach for a different protein system, distinct from PHF10.

- Unfortunately, experiments of our approach testing with isoforms of other proteins are not completed and we cannot include them in this paper. 

Round 2

Reviewer 3 Report

This is a revised version of the manuscript titled "New approach for study of isoforms and high-homology proteins in mammalian cells” submitted by Soshnikova and colleagues. Having carefully reviewed the revised version of the manuscript, I am pleased to acknowledge that the authors have diligently addressed all the drawbacks that were identified in the initial submission. Taken together, all the adjustments and corrections made by the authors have significantly enhanced the manuscript's quality. My recommendations is to publish the manuscript in current form.

English language is fine and only minor corrections are required.